# Experiences and needs of unaccompanied irregular migrant minors who arrive in Spain on small boats: A qualitative study

Ousmane Berthe-Kone[1] , José Granero Molina[1,2] , Cayetano Fernández-Sola[1,2] , María del Mar Jiménez-Lasserrotte[1] and Maria Auxiliadora Robles-Bello[3]

[1]Department of Nursing, Physiotherapy and Medicine, University of Almeria, Almeria, Spain; [2]Faculty of Health Sciences, Autonomous University of Chile, Santiago, Chile and [3]Department of Psychology, University of Jaen, Jaen, Spain

## Research Article

**Keywords:**
migration; unaccompanied minors; qualitative study; health problems; psychosocial problems; irregular migration.

**Correspondence author:**
Maria Auxiliadora Robles-Bello;
Email: marobles@ujaen.es

## Abstract

The European Union receives thousands of unaccompanied irregular migrant minors every year, but little is known about their life experiences during the migration process. The aim of this study is to describe their experiences as minors when they arrived in Spain in small boats, which will help to understand their psychosocial and health needs. A descriptive qualitative study was undertaken. In-depth interviews were conducted with 18 unaccompanied irregular migrants (15 men and 3 women) from different African countries with a mean age of 20.05 years (SD = 2.77). Thematic analysis was used to analyse the data. Three main themes emerged such as (1) unaccompanied irregular migrant minors: risking it all for a better life; (2) redefining your identity as a means of adaptation and (3) obtaining legal status to avoid deportation. Unaccompanied migrant minors risk their lives on the migration journey, but do not always find better conditions in the destination country. The unaccompanied irregular migrant minors are forced to rebuild their lives at a high cost; they experience rejection from the host society and their culture of origin, which has a negative impact on their physical and psychological health over time.

## Impact statement

The findings of this study highlight the profound human cost and severe challenges that unaccompanied irregular migrant minors (UIMM) face during their journey and subsequent integration into Europe. These individuals are driven to flee their home countries to flee from extreme violence, poverty, or a lack of opportunities, often risking their lives in pursuit of a better future. The study highlights the highly vulnerable situation these minors endure, including exposure to physical danger, exploitation and trauma. Despite their resilience, the UIMM face significant obstacles in adapting to their new environment, navigating legal systems and securing stable employment, which impacts their mental and physical well-being. The data presented highlight the need for urgent action to protect and support UIMM, including policies that address their unique vulnerabilities, and promote their successful integration into host societies.

## Introduction

Migration is a common phenomenon in human history, and is currently on the rise (UNHCR 2021). There were 281 million international migrants in 2020, accounting for 3.6% of the world's population (IOM, 2022). Some 1.7 million people crossed international borders in 2021 in search of protection (UNHCR 2021). More than 108 million were displaced by the end of 2022, which rose even further in 2023 (UNHCR 2021; Sánchez-Teruel and Robles-Bello 2022). Central and West Africa have high levels of migration (Kassam et al. 2022). An increase in international migration is associated with political instability (Urrego-Parra et al. 2022), natural disasters, political, social or religious persecution, family conflict or the pursuit of new opportunities (Sánchez-Teruel et al. 2020). Migration flows, a third of which were destined for Europe (IOM 2022; Kassam et al. 2022), reached their peak in the last decade of the 21st century (Agbata et al. 2019) and there was a sharp increase in irregular migration (Mwanri et al. 2022; Silva et al. 2022). Irregular migrants (IMs) are people who enter a country without authorisation and lack legal status to remain in a transit or host country (IOM 2022; Brance et al. 2023). There has been a considerable increase in both the number of female IMs with minors (López-Domene et al. 2019) and in unaccompanied IM minors (UIMM) (ECOR 2018). More than 50% of the world's migrants are minors (IOM 2022), 25% of whom arrive in Europe as UIMM.

Mediterranean countries receive 22.8% of IMs worldwide (Brance et al. 2023). In 2022, 16,718 IMs arrived in Spain, 12,083 of whom entered via the sea (IOM 2022; Ministerio del Interior 2022). Migration poses a risk to human life (WHO 2022). Since 2015, there have been 48,000 migrant deaths worldwide, of which 20,464 occurred in the Mediterranean Sea (Müller et al. 2019). IMs traverse the Sahara Desert to North Africa, and then cross the Mediterranean Sea to Europe (Stevens 2020). IMs risk their lives and endure thirst, hunger, burns and abuse before being rescued at sea and attended to by the Spanish Red Cross (Jiménez-Lasserrotte et al. 2023). Although the majority of IMs are men, the number of women and minors is increasing (IOM 2022). UIMM are an extremely high-risk population (WHO 2022) who have specific care needs at each stage of the migration process (Höhne et al. 2022; Guillot-Wright et al. 2022). UIMM flee gender-based discrimination, violence and social and political conflicts (Henkelmann et al. 2020; Fauk et al. 2021) in search of improved safety, well-being and quality of life (Leku et al. 2022). UIMM are extremely vulnerable physically and psychologically (Verhülsdonk and Molendijk 2021). They are subject to violence, persecution, sexual abuse or imprisonment (Ciaramella et al. 2022). High levels of psychological distress (Vega Potler et al. 2023) are compounded by high rates of mortality and despair (Granero-Molina et al. 2021).

In Spain, after being rescued at sea, UIMM receive emergency care (ECOR 2018) and spend a maximum of 3 days in police cells (Granero-Molina et al. 2022). After osteometric tests, the minors are transferred to Humanitarian Reception Centres (Kassam et al. 2022). UIMM present with chronic conditions, such as asthma or diabetes, and travel-related injuries, such as wounds or burns (Jiménez-Lasserrotte et al. 2023). Algerian and Moroccan UIMM tend to travel alone or in groups, and sub-Saharan Africans often arrive alone (Granero-Molina et al. 2022). After spending time in a Humanitarian Reception Centre, UIMM start the process of integration into the host country, during which they face precarious socio-economic conditions, discrimination, difficulty in accessing health services and a lack of social support (Brance et al. 2023).

UIMM experience difficulties in obtaining legal status and finding employment and suffer from physical and mental health problems (Mwanri et al. 2022). There is also a high incidence of alcohol abuse, pathological gambling, suicidal thoughts and depression among UIMM, who are also at high risk of sexual abuse (Höhne et al. 2022; Mwanri et al. 2022). On top of xenophobia (Sánchez-Teruel et al. 2020; Verhülsdonk and Molendijk 2021), the UIMM also suffer from stress, anxiety, depression and post-traumatic stress disorders (Leku et al. 2022; WHO 2022). Long-term problems associated with labour exploitation were also identified among Filipino and Indonesian immigrants in Chile (Chan and Trahms 2024). Meanwhile, Latino immigrants arriving in the United States exhibited symptoms of psychological trauma as a result of forced separation that persisted even after reunification (Hampton et al. 2021). The problem has been studied on a demographic (Pavlidis and Karakasi 2019; Vega Potler et al. 2023), economic (Ciaramella et al. 2022), social (Guillot-Wright et al. 2022), cultural and health (Kassam et al. 2022; WHO 2022) level, but little is known about how UIMM experience their needs/problems during the migration process.

The aim of this study is therefore to describe the experiences of UIMM who arrive in Spain on small boats with regard to their psychosocial and health needs.

## Design

A descriptive qualitative study was designed. This approach allows for an in-depth description of little-known phenomena, such as the UIMM's experiences, through the exploration of the participants' own experiences (Bradshaw et al. 2017). Consolidated criteria for qualitative research reporting were used (Tong et al. 2007).

## Participants and context

The study was conducted in Almería (Spain) a city in south-eastern Spain between December 2020 and May 2021. Convenience and snowball sampling were used to recruit participants according to the following inclusion criteria: (1) to be a UIMM; (2) to have undertaken a migration journey to Spain on a small boat; (3) to speak Bambara, French, English or Spanish and (4) to currently be of legal age. To contact and recruit the participants, a researcher (OBK), who had volunteered in several UIMM drop-in centres, contacted former UIMM whose contact details he had kept. The researcher invited them to participate in the study by text message and those who responded positively were contacted by telephone to arrange an interview. Initially, 25 UIMM were contacted, but only 18 were willing to participate. Seven UIMM declined to participate due to a lack of trust, fear or unwillingness to recall their traumatic experiences. The final sample consisted of 18 UIMM with a mean age of 20.05 years (SD = 2.77). The socio-demographic characteristics of the participants are shown in the Table 3.

## Data collection

Following approval from the research and ethics committee, the researchers conducted in-depth interviews (IDIs) in meeting rooms at a university in southern Spain that lasted an average of 58 min. The first author contacted the participants by telephone and provided a detailed explanation of the purpose of the study. The UIMM participated voluntarily and prior to the start of the study, the protocol was explained, confidentiality of data was guaranteed, and the informed consent form was signed. Semi-structured IDIs were conducted face-to-face by trained researchers (one researcher was a UIMM) in the presence of cultural mediators who spoke the language of the participants. Socio-demographic data were collected prior to the interview and a script of questions was used (Table 1). IDIs were recorded, transcribed and subsequently included in the hermeneutic unit analysed with the ATLAS-TI 23 software. Data collection ended when saturation had been reached (Green and Thorogood 2018).

## Data analysis

Recordings of the IDIs were transcribed and analysed according to the phases of thematic analysis described by Braun and Clarke (2021): (1) familiarisation with the data: analysis was carried out as data were being collected to gain an early understanding of the participants' experiences; (2) generation of initial codes before grouping them into categories; (3) inductive analysis to generate possible themes; (4) review of themes: two researchers independently reviewed transcripts and results, sub-themes and themes; (5) theme definition and naming: analysis and fine-tuning of the details of each theme and (6) report writing: selecting examples of themes and sub-themes, relating the analysis to the research question and writing the final report. Data analysis (Table 2) was conducted using ATLAS.ti.23.

**Table 1.** Interview script

| Phase | Themes | Content/sample questions |
|---|---|---|
| Introduction | Objective | To describe the experiences of UIMM who arrive in Spain on small boats with regard to their psychosocial and health needs at different stages of the migration process. |
| | Ethical considerations | Inform about voluntary nature of participation, recording, consent, withdrawal and confidentiality. |
| Opening | Opening question | Could you tell me about your experience of emigrating to Spain? |
| Development | Interview script | Tell me about your life in your country with your family.<br>How did the situation affect your health?<br>Tell me about the journey, preparation, overland crossing, waiting in North Africa, the journey by boat.<br>Tell me about your experience in the detention centre.<br>Tell me about the process of social and professional integration in the host country. |
| Closing | Final question | Would you like to add anything else? |
| | Acknowledgements | Thank them for their time and let them know that they can contact us should they wish. |

**Table 2.** Example of data analysis

| Quote | Initial code | Units of meaning | Subtheme | Theme |
|---|---|---|---|---|
| *"I have worked in the agricultural sector. It was a very bad experience especially working for my boss, because we could work in up to 40 degrees in the greenhouse, which is not recommended. Right at noon he would start fumigating while we were working with the excuse that it is environmentally friendly, but I was always against it…."* IDI13 | Need to work, bad experience, effort, exposure to adverse weather conditions. | Coming of age, anxiety, depression, misinformation, language barrier, pessimism, legal status. | Legal status: a bureaucratic labyrinth | Obtaining legal status to avoid deportation. |

IDI = In-depth interview.

**Table 3.** Socio-demographic characteristics of the participants (*N* = 18)

| Participant | Sex | Age | Country | Occupation | Age at what the participants arrived in Spain | How long were they travelling | How long they have been in the host country before the interview |
|---|---|---|---|---|---|---|---|
| IDI1 | Female | 20 | Mali | Agriculture | 16 | 3 years | 2 years |
| IDI2 | Male | 18 | Senegal | Student | 17 | 2 years | 1 year |
| IDI3 | Female | 19 | Guinea | Student | 16 | 1 year | 2 years |
| IDI4 | Male | 21 | Morocco | Agriculture | 14 | 1 year | 4 years |
| IDI5 | Male | 18 | Gambia | Student | 15 | 3 years | 3 years |
| IDI6 | Male | 18 | Senegal | Student | 15 | 1 year | 3 years |
| IDI7 | Male | 18 | Morocco | Student | 16 | 2 months | 2 years |
| IDI8 | Male | 18 | Mali | Student | 15 | 2 years | 3 years |
| IDI9 | Male | 19 | Cameroon | Student | 17 | 6 months | 1 year |
| IDI10 | Male | 19 | Morocco | Agriculture | 16 | 1 month | 2 years |
| IDI11 | Male | 20 | Morocco | Agriculture | 16 | 1 week | 2 years |
| IDI12 | Female | 18 | Algeria | Student | 17 | 2 months | 1 year |
| IDI13 | Male | 19 | Morocco | Agriculture | 17 | 4 days | 1 year |
| IDI14 | Male | 24 | Senegal | Agriculture | 15 | 2 years | 3 years |
| IDI15 | Male | 21 | Guinea | Student | 17 | 3 years | 1 year |
| IDI16 | Male | 18 | Ivory Coast | Student | 17 | 4 years | 1 year |
| IDI17 | Male | 25 | Morocco | Cook | 15 | 3 days | 3 years |
| IDI18 | Male | 28 | Algeria | Carpenter | 16 | 5 months | 2 years |

IDI: In-depth interview.

## Rigour

To ensure rigour, Guba and Lincoln's (1994) criteria were followed. Credibility: the data collection process was detailed, data interpretation was supported by researcher triangulation, and the analysis process was reviewed by two independent reviewers. Transferability: a detailed description of the study setting, participants and method was provided. Reliability: interpretations and descriptions were reviewed by two expert researchers outside the study. Confirmability: this was achieved through participants reviewing the transcripts, confirming the content and verifying the results.

## Ethical considerations

The study was approved by the Department's Ethics and Research Committee (EFM-03/20). The research was conducted in accordance with the ethical principles of the Declaration of Helsinki. The participants were asked for written informed consent and permission to record interviews. All recordings were made in accordance with current legislation on personal data protection.

## Results

All participants were UIMM from African countries ($N$ = 18) (Table 3), of whom 16.7% were female and 83.3% male. The mean age was 20.05 years (SD = 2.77). In terms of their current occupation, 50% are students, 33.3% work in agriculture, 5.5% are cooks, 5.5% are carpenters and 5.5% work in housekeeping. Three themes and eight sub-themes emerged from the inductive analysis of the data, which allowed for a better understanding of the UIMM's experiences (Table 4).

## 1. UIMM: Risking it all for a better life

This theme reflects the situation of thousands of UIMM driven to leave their home countries in search of a better life. Rationalising this decision involves considering the UIMM's human rights, vulnerability and lack of alternatives in their countries of origin, where they face extremely difficult situations of violence, extreme poverty, armed conflict, natural disasters or lack of educational and development opportunities.

### 1.1. The human cost of reaching the promised land

The migration process is an assault on human life. Illegal migration in southern Europe takes place in a highly vulnerable context that particularly affects UIMM. Minors flee their countries of origin due to political crises, famine or lack of prospects in search of a better life and future with more opportunities.

*I considered leaving my country because of the political crisis, chaos, unsafe conditions and lack of opportunities. The only alternative was to leave my country, … but I knew I might die trying* [IDI8].

If the UIMM is female, the risk is greater; as well as the aforementioned reasons, girls flee their countries of origin due to forced marriages that take place in order to uphold agreements between families or to pay off debts.

*My Dad died and he had debts. They wanted to force me to marry an older man, I was only 16 years old. I didn't want to, my mother begged me for forgiveness and begged me not to run away* [IDI1].

The participants were informed of the difficulty and risks of the journey. The UIMM were still minors at the time and described the migration journey as "playing with death." They were aware of the hardship that awaited them, including the possibility of death. UIMM are extremely vulnerable; they travel alone, are exposed to extreme weather conditions, and suffer from severe dehydration and malnutrition. They are also at high risk of falling into the hands of mafias or being sexually abused. During the migration process, they are hungry, sleep on the streets and travel on foot or in trucks, hidden in suitcases. This is how one participant described it:

*The suffering began when we crossed the border of Mali and Algeria. In the Sahara Desert, it is hellishly hot during the day and deathly cold at night. The journey lasted about 24 hours, with no food, but the problem was water. We were carrying four bottles of water for eight people. It was hell, I thought we were going to die. I try not to think about all this, but the memory is still there* [IDI5].

All the participants in the study agreed that the route north is very dangerous; they can get sick, they can be stopped by the police in any country they pass through, and they are highly vulnerable to sexual abuse. Nobody asks or cares about them. In the middle of the desert they can be assaulted by Tuareg gangs, who take all their belongings, assault the men and then let them go on. During the journey, they have no one to protect them. If they disappear, no one looks for them, and they are not even a number because only the

**Table 4.** Themes, sub-themes and units of meaning

| Themes | Sub-themes | Units of meaning |
|---|---|---|
| 1. UIMM: risking it all for a better life | 1.1 The human cost of reaching the promised land | Political crisis, surviving adversity, violence, lack of rights, difficult childhoods, unemployment, poverty, new opportunities. |
| | 1.2 From despair to salvation | Malnutrition and dehydration, fear, journey in a small boat, near-death experience, detention, settlements, police control. |
| 2. Redefining your identity as a means of adaptation | 2.1 The uncertainty of redefining your life | Help to integrate, learning the language, work dictates, dealing with papers, suffering, cohabitation, lack of support. |
| | 2.2 Social stigma and finding refuge among peers | Discrimination, xenophobia, difficulty of integration, grouping by race, socio-economic problems. |
| 3. Obtaining legal status to avoid deportation | 3.1 Legal status: a bureaucratic labyrinth | Coming of age, anxiety, depression, misinformation, language barrier, pessimism, legal status, working legally, will to live, composure, self-esteem, solidarity. |
| | 3.2 The vicious circle of job seeking | Effort, work dictates life, building a future, fulfilling a mission, labour exploitation, lack of dignity, lack of rest, contempt and fear. |

UIMM = Unaccompanied irregular migrant minors.

dead are counted. When they arrive in Europe, they eventually become aware of the dangers they have overcome.

*In the middle of the Sahara Desert, an armed group [of Tuareg people] stopped our car, took our water and money, and beat us up* [IDI5].

*The border between Algeria and Morocco is another living hell, a clandestine route of more than 20 kilometres that is crossed on foot. My ankle buckled, I had to travel with a swollen ankle, in pain. But you can't stay behind, you can get lost, be caught by the border police, imprisoned, tortured and … returned to the desert to your fate* [IDI14].

Before embarking on the boat journey, IMs wait in settlements in the Moroccan mountains. The UIMM described this period as dangerous and inhumane; they were forced to sleep rough, surrounded by adult IMs and to survive as best they could. They were subjected to searches, robberies, violence and no minimum standards of food, hygiene or health. This is how one participant described it:

*We lived in the mountains like animals, there was no water, and in the settlement, I didn't even get to take a shower. It was very hot during the day and cold at night. When it rained there was an unbearable smell. People fought each other for a piece of hard bread and as for the minors … much worse, we didn't eat for days* [IDI6].

The waiting time for departure to Spain is variable. While waiting, they are at the mercy of mafias and raids by the Moroccan police. As UIMM, they are easy targets; they have no protection from family or other adults, which makes them even more vulnerable.

*They tricked me, they drove me to the sea, you have to pay … and when you get there, they leave you stranded. The police caught me three times. They take your money and mobile phone, they put you in a truck to a prison for three days and then they release you in the south, or as far as Marrakech, … and you have to come back and do the whole journey again* [IDI5].

The participants recalled their time in these settlements as long and unbearable. They did not have their basic health needs covered; this included insufficient hydration, nutrition, safety and rest. In addition, they suffered abuse and persecution from Moroccan gangs at night. They lived under constant stress, which caused problems, such as diarrhoea, malnutrition, headaches, insomnia, fear and anxiety.

*One night they attacked us, burnt everything, chased us and arrested some people. I hid between two big rocks and stayed there until the sun came up. A few days later they said that some people had been imprisoned and to this day I haven't had any news about them. … I think about it and I feel both sad and lucky to have survived it* [IDI13].

Reaching Europe is the dream that fuels their hope for a better life. Most of the participants described their longing to reach Europe as a way of escaping their personal situation, poverty, war or a lack of freedom in their country. However, this dream turned sour when they recalled the violence, loneliness or the long wait they had endured.

*The day I arrived in Morocco I thought I was so close that I couldn't let anything stand in my way. I never thought I would be there for three years. I felt like a prisoner. Time goes by very slowly in those mountains when you're lying there like a dog. And it was no better when we went to the city, people didn't even look us in the face … we were invisible* [IDI14].

The UIMM clung to the idea of surviving and moving on despite the difficulties they faced and the people who got in their way. They held on to the hope that the following day could not get any worse and that they had already overcome the worst experience in Africa.

However, new challenges led their situation to go from bad to worse. They focused on survival, and as no one was there to help or defend them, they were forced to support each other. As one participant said, the UIMM were the last to get into the boat to cross the sea, they were in the worst places, they hardly got water or food, and they were splashed with petrol, causing burns on their skin. At no point did they lose their resilience, but they did think that their lives could end at any time. As one participant put it:

*I never thought that that plastic boat would reach dry land… I thought I would drown in the sea. I couldn't think of anything, my body didn't respond, I started to feel dizzy and I begged to get to Spain! … I only remember thinking of my mother, that I would never see my mother again* [IDI15].

The participants described the hardship of crossing the sea by boat. They spent many hours swaying back and forth in the waves of the sea, exposed to the sun and unable to see the horizon. Most of them had never seen the sea and did not know how to swim, which led to fear, anxiety, vomiting, hypothermia, dehydration and burns. The UIMM described it as a traumatic experience that they never wanted to relive.

*They put us on the boat heading for Spain at 20:00 with some bottles of water and biscuits. I thought that the suffering was over, but when the waves started to hit us, I was so scared that I no longer felt cold but rather anguish, rage and sadness… I couldn't take it anymore and I couldn't stop vomiting* [IDI18].

*I left from Nouadhibou in a boat with 44 people and it took us four days to reach land. We took food and drink with us, but it's an awful experience. There are waves, people vomit, there is a bad smell, … I was stuck, I couldn't even move my legs!* [IDI17].

## 1.2. From despair to salvation

Like other European countries, Spain is subject to international regulations that establish rights for UIMM and asylum seekers. When UIMM arrive on the coast, emergency care protocols are activated, which focus on healthcare, ensure their well-being and meet their health needs. After being intercepted by the *Guardia Civil* (Spanish police force) or Red Cross at sea and arriving ashore, a general health assessment is carried out, including a medical examination, vaccinations and treatment of previous illnesses or injuries. In addition to this, they are provided with food, water, blankets or clothes.

*We were lost at sea, without hope… suddenly we heard a helicopter and minutes later a Red Cross boat appeared and picked us up. Everyone wanted to drink water, but we were told that after four days without food or water it had to be little by little. When we arrived at the port, the doctor asked us if we had drunk sea water* [IDI18].

Unlike irregular adult migrants, mainly from North Africa or from countries with extradition agreements, UIMM seek to be detained in order to be protected by the state. When they reach the coast, they jump into the sea and swim to the shore or wait near the shore to be picked up by the police. UIMM are provided with initial healthcare separately from adults. Emergency care covers treatment of injuries, burns, fever, wounds or dermatological diseases (scabies). Information regarding their identity and country of origin is also requested. Upon arrival, they are exhausted and stare blankly into space.

*After disembarking at the port, they started asking us questions, … I didn't understand anything. I was so weak that I couldn't even speak; they started to examine us, and they put a red bracelet on me* [IDI12].

Other IMs arrive in a very bad state of health and are taken to hospital. As one participant who lost consciousness recalled:

*I don't remember anything. We ran out of water, food and petrol on the boat. I started drinking sea water and lost consciousness. When I woke up, I was lying on a bed and I saw nurses and two policemen talking to me in Spanish. I was scared to death!!* [IDI16].

Prioritising emergency care for UIMM includes providing them with water, food, showers and clean clothes. Those suspected of being adults are then transferred to the hospital for a bone assessment prior to being admitted to the reception centre. It is essential to determine whether UIMM are minors, as this determines their legal status and protective measures. In some cases, the age of the UIMM can be questionable so medical examinations and interviews are conducted to establish their age.

*They gave us water, juice, muffins and then took us to the shower. After the shower, they treated the burn on my neck and thigh that I got from the sea water and sun exposure. Then they took me to the hospital, they took my blood, did x-rays and put me on a saline drip… I spent a night in the hospital and the next day they took me to the centre for minors* [IDI2].

When they are confirmed as minors, UIMM are placed in a Humanitarian Reception Centre where they receive accommodation, medical care, education and psychosocial support. They have access to basic and specialised medical care, mental health services and psychosocial support. In the face of emotional and psychological challenges, social workers and psychologists provide them with support to adapt to their new situation and address their trauma. As one participant said, it is the point at which they felt their life was beginning to change.

*My life turned around when I was admitted to the centre. It was very difficult, but with the support of the educators, I was gradually integrated into my new life. One day they took us to the doctor for blood tests and a Mantoux test, and the following week we went back for vaccinations. Every week we had an appointment with the centre's psychologist … although I didn't understand him very well because I didn't speak Spanish* [IDI3].

## 2. Redefining your identity as a means of adaptation

The participants detailed all the suffering they experienced in the small boats on the journey to the Spanish coast, and later described the long process of uncertainty that comes with being in an unknown country. Everything is different for them in every way, from the language to education, social norms and socio-economic problems. Finding one's place in the host society and establishing social relations in a cultural context that is different from the country of origin involves redefining one's own identity and acquiring a great capacity for adaptation and resilience.

### 2.1. The uncertainty of redefining your life
This section focuses on the participants' experiences of arriving in Spain and being faced with a different sociocultural context from that of their home countries. It was not easy for the UIMM to adapt to the reception centre and to Spanish society; the participants felt lost, disoriented and totally uncertain about their future. They were faced with the major obstacles of language barriers, social problems between minors of different ages and origins, and a lack of social and family support, which led to fear, stress and constant anxiety. As one participant said, they needed someone to guide them through their first contact with the host society. They needed support from people who spoke their language in order to improve social interaction, integration and their ability to manage everyday situations.

*The language is a barrier, a very big problem, I thought people here spoke English, but they don't … and that's why there are so many misunderstandings. Once they took 7 test tubes, stool, blood, urine … what for? … I didn't understand anything they explained to me!* [IDI1].

The lack of family support made the UIMM feel uncertain about how to navigate their lives. They did not know how the centre worked or what was expected of them. They lived 1 day at a time, wondering about their future. Adapting to the centre and to the host society was not easy and was a gradual process. The culture shock when interacting with the centre's staff and other minors produced stress, anxiety, feelings of loss of identity and even rejection.

*I didn't know it was a centre for minors. The anxiety on arrival, not knowing where it was, not understanding the language… At the beginning it was very difficult. I felt locked up, there were many rules and I wanted to leave, to go to France with my brother… but they told me that it was not easy, that they must do DNA tests and a lot of paperwork, that it would take at least 6 months! I was living between two realities. In the centre I was eating different food, they did not speak my mother tongue, they did not pronounce my name properly, … my life would never be the same again!* [IDI18].

### 2.2. Social stigma and seeking refuge among peers
Social rejection is a fundamental factor in the UIMM's integration process in the host country. They deal with problems related to violence, poverty and vulnerability. Furthermore, the long legalisation process, the lack of expectations and lack of information about their future generate feelings of stress, anxiety and uncertainty. The UIMM feel discriminated against because of their race, religion, nationality or social group; they feel rejected and associated with situations of violence, poverty and vulnerability. This triggers negative feelings, leading them to bond with people in similar conditions. The lengthy legalisation process and lack of information increases their feelings of uncertainty. They do not know what the future holds for them, so they cling to each other.

*I was in a class where I was the only black African boy, … yes, I was discriminated against by other classmates. I didn't understand what they were saying to me, and they didn't understand me. We were constantly confronting one another, and I was often punished without knowing why…* [IDI5].

During the integration process, the UIMM experience discrimination when using transport, training or accessing leisure facilities, where they may even be denied access because of their colour or race. The participants felt that their personal circumstances did not align with the host country.

*On the street, in some public places, I have experienced racism and very offensive comments. I'm still experiencing that, and it makes me more and more enraged with some of the people here* [IDI7].

## 3. Obtaining legal status to avoid deportation

One of the biggest obstacles the UIMM face is legalising their administrative situation in the host country, which results in a precarious employment situation and poor quality of life. Not having documentation makes them more vulnerable to social inequality and legal issues.

### 3.1. Legal status: A bureaucratic labyrinth
The bureaucratic labyrinth refers to the procedural barriers that hinder the UIMM's process of administrative legalisation. Their lack of information, computer literacy and language competence

force them to ask for help, which makes them aware of how dependent they are. This is how one participant put it:

> People coming from abroad are not only foreigners, but they also find it difficult to understand the systems in place to regularise their status, to apply for a residence permit or refugee status, … all of this is very complicated and difficult to understand [IDI7].

Not having legal papers is an obstacle that exacerbates their work conditions, making them more susceptible to exploitation, social inequality and poverty. When they arrived in Spain, they did not know the system and felt defenceless, but this situation has not changed over time. For example, they did not know if they could go to the doctor if they were sick, if they could ask for a day off from work, or even if they were allowed to go to work sick. All these concerns made them anxious, and the wait to legalise their status in the host country seemed endless.

> What matters is having papers, to have legal status…. that's why we ask for "papers, papers", so that we don't have to go back" [IDI11].

When they reached legal age, the UIMM had to leave the Humanitarian Reception Centre and move into supervised flats. Many of them had not had time for training, as a boy from Gambia recounted:

> When I turned 18, I was transferred from the centre for minors to a flat for UIMM. I had no residency papers, I spoke almost no Spanish, I had no education … I argued with a flatmate and was kicked out … I found myself on the street again. I lived through all that suffering again with no one to give me a hug!! [IDI15].

### 3.2. The vicious circle of job seeking

The UIMM-related work to personal identity; they need a job to feel fulfilled and integrated in the society of the host country. However, they perceived that being an UIMM reduced their chances of finding a decent job. The participants found themselves in a vicious circle. On the one hand, the law does not allow people with an irregular status to be employed, so they could not get a work contract. Yet, on the other hand, in order to get a residence permit, they need to have had a work contract for at least 1 year. This pushes UIMM into exploitative labour, long working hours in agriculture or housekeeping, with no breaks and low pay. This is how these participants expressed the situation:

> I have worked in the agricultural sector. It was a very bad experience especially working for my boss, because we could work in up to 40 degrees in the greenhouse, which is not recommended. Right at noon he would start fumigating while we were working with the excuse that it is environmentally friendly, but I was always against it [IDI13].

> You need papers to work. I wanted to sort out my administrative situation, but my asylum application was rejected. Without work or a residence permit, I have only managed to find a job with a lady who pays me € 5 an hour [IDI11].

The female participants perceive that their race and gender (their status as women and black), exacerbates their discrimination, which results in poorer financial and employment conditions. The immigrant population feels condemned to precarious jobs, labour exploitation, harsh weather conditions, a lack of rest and fear of dismissal. For the UIMM, work is not only a source of income but also their key to feeling part of society. When they recalled periods of unemployment, they felt devalued, frustrated and demotivated. At other times, as one participant stated, their work conditions did not allow for dignified living conditions.

> Once you get here, you realise that it's not enough to survive the journey. You are a black woman, and that means that racism and discrimination continue. The jobs I am offered are the hardest, and they pay me the minimum. They tell me I should be grateful for anything… [IDI1].

> Without papers you have no rights! You are paid below minimum wage, it's impossible to get paid overtime … or you get paid less. When there are no papers, the boss sets the price! You can't do anything, you are at their mercy! [IDI4].

## Discussion

The aim of this study was to describe the experiences of UIMM who arrive in Spain on small boats with regard to their psychosocial and health needs. Key factors that contribute to their decision to migrate are the desire for a better life, hope of reaching Europe, search for better financial opportunities, flight from violence/discrimination and the aspiration for a free and prosperous life (Ciaramella et al. 2022). The text discusses the challenges faced by (UIMM) who risk everything to flee from violence, poverty and lack of opportunities in their home countries, often enduring a perilous journey to Europe. Upon arriving in Spain, they encounter a long, stressful process of adapting to a new culture and navigating complex legal and social systems, which often leads to feelings of disorientation and anxiety. Their lack of legal status exacerbates their vulnerability, trapping them in cycles of exploitation and social exclusion. Their identity and future prospects are deeply influenced by these intersecting challenges of race, gender and legal uncertainty, making their integration into society difficult and fraught with systemic inequalities.

The participants told us that the migration process is fraught with difficulties, in line with other studies (Leku et al. 2022); UIMM face situations of violence, loneliness, discrimination and a lack of resources during their journey, which has been documented in previous research (Granero-Molina et al. 2021; Sánchez-Teruel et al. 2022). This causes them emotional and physical distress (Henkelmann et al. 2020; Verhülsdonk et al. 2021), together with a feeling of marginalisation and invisibility (López-Domene et al. 2019). From a professional point of view, this study teaches us that it is necessary to address the underlying causes of forced migration among UIMM, such as poverty, violence and a lack of opportunities in their countries of origin (WHO 2022). This involves addressing structural problems and working with the origin, transit and destination countries to provide safe and legal migration pathways (UNHCR 2021). Consistent with other studies (Mwanri et al. 2022; Silva et al. 2022), when UIMM arrive in a new country they experience a sense of loss, disorientation and unfamiliarity with the culture, language and social norms. This leads to anxiety and feelings of uncertainty. UIMM ask for support to guide them in the first phases of contact with the host society, in order to facilitate integration (Ciaramella et al. 2022). In line with other studies (UNHCR 2021; Leku et al. 2022), social and family support is key in the UIMM's adaptation process, therefore more support resources are needed from the time of arrival and through the adaptation process (WHO 2022).

As reported in a recent review (Ciaramella et al. 2022), the UIMM of this study also experience rejection and discrimination based on their ethnic, cultural or racial origin. Social stigma leads to feelings of exclusion, forcing them to seek refuge with similar people who offer them reassurance and understanding. The discrimination experienced by UIMM can have a negative impact on their psychological and social well-being, resulting in stress,

anxiety, depression and decreased quality of life (Verhülsdonk et al. 2021). As reported in other studies, language barriers hinder communication, integration, job seeking and leisure activities for UIMM (Rousseau and Frounfelker 2019). From a professional point of view it is necessary to promote intercultural education and raising awareness in the host society is essential to reduce stigma and discrimination against UIMM (UNHCR 2021; WHO 2022).

The participants highlighted bureaucratic and legal obstacles when trying to obtain legal status in a new country. As previous research (Loayza-Rivas and Fernández Castro 2020) shows, these obstacles include the complexity of administrative procedures, lack of information, dependence on third parties, legal uncertainty and long waiting times. Facing an unfamiliar and complex system leads to feelings of lost autonomy, distress and depression among UIMM (UNHCR 2021; Leku et al. 2022); simplifying administrative procedures and improving access to information is essential (Sánchez-Teruel and Robles-Bello 2022). According to this study's UIMM, differences in race, nationality or ethnicity reduce their employment opportunities and increase their vulnerability to labour exploitation. We concur with other research that xenophobia is a barrier for UIMM in accessing decent jobs (Lam et al. 2019). We also acknowledge the precarious working conditions faced by migrants (Granero-Molina et al. 2021); long working hours, a lack of rest and financial hardship have a negative impact on their physical and mental health (Henkelmann et al. 2020; Verhülsdonk et al. 2021). The experiences of our participants corroborate the need for labour protection for UIMM to ensure fair and safe work conditions, which in turn is key for them to experience a feeling of social integration (WHO 2022).

From a professional point of view, it is important to emphasise the importance of ensuring that unaccompanied minors receive adequate support and accompaniment throughout the migration process to safeguard their well-being. Research has consistently shown that the presence of a supportive adult or guardian significantly mitigates the numerous risks faced by these vulnerable individuals (Solberg et al. 2020). Studies highlight that accompanied minors experience lower levels of trauma, better access to essential services and improved psychological outcomes compared to those who migrate alone. The presence of a responsible adult can provide critical emotional support, protection from exploitation and guidance through complex legal and social systems (Brun-Rambaud et al. 2023). This support not only enhances their chances of successfully integrating into the host society but also helps in addressing the trauma and instability experienced during their migration journey. Thus, ensuring that unaccompanied minors have access to appropriate guardianship and support mechanisms is fundamental in promoting their safety, health and overall well-being (Höhne et al. 2023).

## Limitations

The limitations of the study relate to the socio-demographic characteristics of the participants. Although it is a heterogeneous sample with participants from different African countries, future research could include UIMM who live in different regions of Spain. The study could also have delved more deeply into cultural and religious norms, which could have influenced their experiences. Regarding sex, the majority of the participants are men; the proportion of female participants could have been increased. Furthermore, if the interviews had taken place at a later date, this could have allowed the UIMM to elaborate further on their experiences, thus enriching

the data. The fact that one of the study's researchers was a UIMM may have encouraged the participants to share their experiences. However, experiences might differ depending on the country of origin. Finally, it is clear that the sample size could be improved, especially with regard to women, which should be taken into account for future research. At the time of the interviews, all of the participants, who arrived as UIMM, were still in Spain. However, we do not know whether their original objective was to reach Spain or another country. Nor do we know whether they had a social support network or family members in Spain. Information on these aspects would have helped us in the interpretation of the results.

## Conclusion

UIMM start the migration process fleeing war, family conflicts and a lack of opportunities, while pursuing the dream of a better life. The migration culture in their countries of origin compels them to risk their lives on a dangerous journey to Europe. Given that they are minors, travel alone, and lack any family or social support, UIMM are at higher risk than adults. Upon arrival in the promised land, they are the first to receive emergency care for their physical, psychological and social needs. Once in Spain, the integration process is hindered by misinformation, lack of knowledge of the culture and customs of the destination country, language barriers, legal uncertainty and fear of deportation. This culture shock pushes them to stick together to face social, work and integration challenges as a group. This situation can negatively affect their physical and psychological health, which only worsens over time. The participants highlighted the need to adapt and redefine their identity in a new cultural context, as well as the legal obstacles that hinder their integration and their access to employment and a dignified life.

**Open peer review.** To view the open peer review materials for this article, please visit http://doi.org/10.1017/gmh.2024.146.

**Data availability statement.** Data presented in this study are available upon request from the corresponding author. Data are not publicly available due to privacy and ethical issues.

**Acknowledgements.** The authors thank all the participants for participating and sharing their experiences. Thanks to the University of Almeria's Health Science Research Group (CTS-451) for their support. Funding for open acces charge: University of Jaen (Spain).

**Competing interest.** The authors declare no conflict of interest.

**Institutional review board statement.** Approved by the Department's Research and Ethics Committee (EFM-03/20). The research was conducted in accordance with the ethical principles of the Declaration of Helsinki.

**Informed consent statement.** Informed consent was obtained from all subjects involved in the study.

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
