## [Reviewer Report]

I am grateful to the editor of the journal and the authors of this manuscript for giving me the opportunity to make some suggestions for improvement:

1. Describe in the abstract the basic demographics of the participants: sex (% female, male or other) and age (mean and standard deviation).

2. Please modify the acronyms MENA or UIMC throughout the manuscript: its correct meaning is Unaccompanied Foreign Minors-UFM or Unaccompanied Immigrant Children (UIC)

3. In the introduction, it is not a good practice to use several references for the same idea, since the authors of said references may offer different conclusions. Check that all the authors included in a sentence say the same thing.

4. Check all references in the text, as some are missing from the reference list. On the other hand, include updated references in the manuscript (no older than 5 years). Older references may only be considered when they are classic definitions of variables or particularly significant authors.

5. In the discussion, try to give your professional opinion to the results obtained, and do not abuse the refutation or confirmation of results from other authors. This is the most interesting part of the discussion.

---

## [Reviewer Report]

The main objective of this study is to use a qualitative methodology to describe and understand the process experienced by minors who decide to emigrate to another country in an irregular and unaccompanied manner. The conditions of special vulnerability of this population highlight the importance of studies that focus on this objective.

The different sections of the work (review of the available information and establishment of objectives, design, results obtained and discussion of the results) have been developed with conceptual and methodological rigour, debating important aspects in order to understand the factors and conditions that encourage these children to make their decision and the different difficulties faced both during the journey and once they arrive in the host country. The information obtained, although it can be anticipated intuitively, suggests areas of work that would limit decision-making in such an uncertain and dangerous journey, such as reception strategies in the country of arrival that improve the well-being and integration of these minors. In this sense, the information offered is relevant and rigorous.

Some aspects that should be complemented in future studies have already been identified by the authors, especially two: the need to integrate a gender perspective, involving a greater number of women among the participants, which has not been possible in this study, and the use of different migratory routes. In addition to these circumstances, I would also like to know more extensively the perspective of minors once they have made the journey; that is, to know their assessment of the decision taken, discrepancies in expectations and personal suggestions for the improvement of the reception process. I would also like to know their assessment of the number of participants and ways to increase it.

As minor changes, I suggest revising the bibliographical references. For example, some references are missing (Brance & Bentall, 2022; Jiménez-Lasserrotte et al., 2023) or incomplete (von et al., 2019) as well as revising the alphabetical order.

---

## [Reviewer Report]

This is an original study about the experience of subjects who migrated irregularly to Spain through a small boat when they were under 18 years of age. Despite the high number of immigrants who arrived to Spain in the last two decades, there are few studies which study the characteristics of this population. It is an interesting topic, but the manuscript needs some changes to solve some problems. Specifically, the following issues need to be addressed:

Title

I suggest to modify the term “children” for “adolescents” or “minors”. It’s not clear from the manuscript that the participants in the study arrived to Spain before 13 years, the usual age to be considered a child.

Abstract

I would recommend supressing the words “and understand” in the abstract and text (introduction, discussion) in the aim of the study.

Impact statement

This statement should be adjusted to the results of the study. It’s explained that the minors received physical and mental health care when they arrived and live in the Humanitarian Reception Centres. Thus, it seems is when they are older, after 18 years of age, when the problems that the authors explained, emerge.

Introduction

The authors state that “Irregular migration poses a risk to human life”, but irregular migrants could arrive by plane or other transportation. “Irregular” is referred to a legal status, as the authors explained earlier in the text. Thus, we recommend to modify through the introduction and discussion this aspect of irregularity and change the term and acronym IM for another more appropriate.

The third paragraph would be started as: “In Spain, after being…” to clarify that this specific process takes place in this country. In the same paragraph, information about minors and adults is mixed. Please, organize it and separate the studies from the two populations. Also in this paragraph, it would be recommended to clarify if the incidence of psychopathology in UIMC are observed in European countries or other areas such as USA, for example. Moreover, it would be needed to clarify the relationship between xenophobia and stress, anxiety, depression and posttraumatic stress disorders.

Participants and context

It would be interesting to specify the city of Spain were the study was conducted. Also, it would be important to detail how the participants were recruited.

It is recommended to move the sociodemographic characteristics of the sample to the Results section.

Results

To better understand the data, it’s important to add information about the age at what the participants arrived in Spain, how much time they were travelling since they left their home and arrived in Spain as well as how many years they have been in the host country when the interview took place.

Also, if the authors could explain, if they have the information, if the goal of the participants were arrived and live in Spain or if they would like to go to another final host country as ID18 declares. Moreover, if they had any relative in Spain, it would be interesting to know.

In the point 3.3 the authors associate the legal problems of the UIMC with the mental health and physical problems and it is recommended to eliminate the sentence, because it is not the result of their study.

In the point 3.3.1 it is not clear in the sentence “For example, they did not know if they could go to the doctor…” if they are talking when they were minors or adults.

In 3.3.2 it is stated that “Social and economic inequality is linked to race and gender…” but this is not associated to the context of the quotes, which are related to work conditions.

Discussion

It’s recommended to start this section with a summary of the results of the study.

It’s not clear enough in this section to know the points related to the results of the current study or the previous studies, it’s a little bit confusing, and needs to be addressed.

It would be important to add a paragraph in the discussion about the importance for the well-being of minors to be accompanied in the process of migration, as has been pointed out in different studies.

Limitations

It would be important to mention the sample size in first place. It’s a significant limitation that is not included.

References

Some of the references are repeated, please check and change it.

Tables

Table 3, the quote of ID11 it’s not clear enough if it is related to the work in his/her native country or in the host country. In the text, this quote is indicated to be said by ID13, review it, please.

---

## [Reviewer Report]

GMH-2024-0099R1

The manuscript has improved with the changes made by the authors. However, they have not addressed and resolved all of the recommendations that this reviewer previously pointed out. The comments below outline issues that still need to be resolved and hopefully will help the authors further improve this article:

Impact statement

This statement should be adjusted to more closely fit the results of the study. The sentence: “These individuals are forced to flee their home countries due to extreme violence, poverty, or lack of opportunities, often risking their lives in pursuit of a better future,” is too strongly worded to be placed in the impact statement. Migrants choose to leave their countries, while other people from the same places do not migrate. Thus, the wording “are forced to flee their home” here seems to be an overstatement, and nothing in the rest of the manuscript attempts to clarify or justify the use of these broad assertions to describe the experience of the subjects in this study.

Introduction

In the sentence: “More than 50% of the world’s migrants are minors and adolescents (IOM, 2022), the concept “minors” includes adolescents. It means below 18 years of age. Thus, the words “and adolescents” should be deleted from this sentence and in similar cases throughout the text.

The paragraph starting with “In Spain, …” seems to mix information from Spanish studies and others from around the world. Please clarify this in the text. Also, it would be good to have three separate paragraphs for the information regarding the process in Spain, information from other countries and the aim of this study.

In this paragraph, it would also be recommendable to offer information about whether the incidence of psychopathology in UMN observed here have been observed in other European countries or in other economically comparable regions such as the USA.

Participants and context

It would be important to detail how the participants were contacted and recruited.

Results

If there were 3 female participants out of 18 participants in the study, the percentage is 16.7%, not 27.7%. Please, change this, and review the percentages.

Also, if the authors have this information, it would be very interesting to know the participants’ original destinations. That is, was their original goal to arrive in Spain and live there, or was it to go to another final host country as the statements of ID18 indicated. Similarly, it would be interesting to know whether these young migrants had any relatives or other social network in Spain before immigrating.

In point 3.1.1. the authors stated: “In the middle of the desert, they can be assaulted by Tuareg gangs, who take all their belongings, separate women and men, assault the men, rape the women, and then let them go on.”, but this did not happen to any of the participants. Please, specify this in the text and add references for this information. In the same point, there is the declaration in cursive of IDI9 (“My Dad died and he had debts. They wanted to force me to marry an older man...”), but according to Table 3 IDI9 is a male, and not a female. Please, review this.

In point 3.1.2 it would be good to clarify what “Guardia Civil” is.

In 3.3.2, there is the statement “Social and economic inequality is linked to race and gender…” but this is not associated with the context of the quotes, which are related to work conditions.

Discussion

The acronym UMN is explained at the beginning of the manuscript. There is no need to describe it here again.

According to the results explained in the manuscript, the sentence: “These minors, especially girls, are highly vulnerable to exploitation, abuse, and extreme hardships during migration.” is not appropriate in the discussion, because, as written, it seems to be a result of your study, when in fact none of the three girls included in it explained these sorts of experiences.

There needs to be more clarity regarding which points relate to the results of the current study, and which relate to previous studies. As currently written, it is a bit confusing, and this needs to be improved.

In the new paragraph added to the discussion about the importance for the well-being of minors that they be accompanied in the process of migration, references would be needed.

Tables

The quote attributed to IDI1 in the new Table 2, is still, attributed to ID13 in the subheading of section 3.3.2 in the text of the article. Please review this.

---

## [Editor Report]

I hope this message finds you well. On behalf of the editorial team, I would like to thank you for your recent revisions to the manuscript. We greatly appreciate the thoughtful updates, and I believe these revisions will significantly enhance the work.

In reviewing the manuscript, we have noted the feedback provided by Reviewer 3 and believe the following points would further benefit from your consideration:

The manuscript has improved with the changes made by the authors. However, they have not addressed and resolved all of the recommendations that this reviewer previously pointed out. The comments below outline issues that still need to be resolved and hopefully will help the authors further improve this article:

Impact statement

This statement should be adjusted to more closely fit the results of the study. The sentence: “These individuals are forced to flee their home countries due to extreme violence, poverty, or lack of opportunities, often risking their lives in pursuit of a better future,” is too strongly worded to be placed in the impact statement. Migrants choose to leave their countries, while other people from the same places do not migrate. Thus, the wording “are forced to flee their home” here seems to be an overstatement, and nothing in the rest of the manuscript attempts to clarify or justify the use of these broad assertions to describe the experience of the subjects in this study.

Introduction

In the sentence: “More than 50% of the world’s migrants are minors and adolescents (IOM, 2022), the concept “minors” includes adolescents. It means below 18 years of age. Thus, the words “and adolescents” should be deleted from this sentence and in similar cases throughout the text.

The paragraph starting with “In Spain, …” seems to mix information from Spanish studies and others from around the world. Please clarify this in the text. Also, it would be good to have three separate paragraphs for the information regarding the process in Spain, information from other countries and the aim of this study.

In this paragraph, it would also be recommendable to offer information about whether the incidence of psychopathology in UMN observed here have been observed in other European countries or in other economically comparable regions such as the USA.

Participants and context

It would be important to detail how the participants were contacted and recruited.

Results

If there were 3 female participants out of 18 participants in the study, the percentage is 16.7%, not 27.7%. Please, change this, and review the percentages.

Also, if the authors have this information, it would be very interesting to know the participants’ original destinations. That is, was their original goal to arrive in Spain and live there, or was it to go to another final host country as the statements of ID18 indicated. Similarly, it would be interesting to know whether these young migrants had any relatives or other social network in Spain before immigrating.

In point 3.1.1. the authors stated: “In the middle of the desert, they can be assaulted by Tuareg gangs, who take all their belongings, separate women and men, assault the men, rape the women, and then let them go on.”, but this did not happen to any of the participants. Please, specify this in the text and add references for this information. In the same point, there is the declaration in cursive of IDI9 (“My Dad died and he had debts. They wanted to force me to marry an older man...”), but according to Table 3 IDI9 is a male, and not a female. Please, review this.

In point 3.1.2 it would be good to clarify what “Guardia Civil” is.

In 3.3.2, there is the statement “Social and economic inequality is linked to race and gender…” but this is not associated with the context of the quotes, which are related to work conditions.

Discussion

The acronym UMN is explained at the beginning of the manuscript. There is no need to describe it here again.

According to the results explained in the manuscript, the sentence: “These minors, especially girls, are highly vulnerable to exploitation, abuse, and extreme hardships during migration.” is not appropriate in the discussion, because, as written, it seems to be a result of your study, when in fact none of the three girls included in it explained these sorts of experiences.

There needs to be more clarity regarding which points relate to the results of the current study, and which relate to previous studies. As currently written, it is a bit confusing, and this needs to be improved.

In the new paragraph added to the discussion about the importance for the well-being of minors that they be accompanied in the process of migration, references would be needed.

Tables

The quote attributed to IDI1 in the new Table 2, is still, attributed to ID13 in the subheading of section 3.3.2 in the text of the article. Please review this.

We are confident that your continued efforts will strengthen the overall contribution of the manuscript, and we are happy to support you in addressing these final points. Please let us know if you have any questions or need further clarification.

We look forward to receiving the revised version and are excited about the potential of your work.